# Mechanisms of Communication in the Mammalian Circadian Timing System

**DOI:** 10.3390/ijms20020343

**Published:** 2019-01-15

**Authors:** Mariana Astiz, Isabel Heyde, Henrik Oster

**Affiliations:** Institute of Neurobiology, Center of Brain, Behavior and Metabolism, University of Lübeck, Marie-Curie-Straße, 23562 Lübeck, Germany; m.astiz@uni-luebeck.de (M.A.); isabel.heyde@live.de (I.H.)

**Keywords:** circadian clocks, peripheral clocks, suprachiasmatic nucleus (SCN), entrainment, resetting, zeitgeber

## Abstract

24-h rhythms in physiology and behaviour are organized by a body-wide network of endogenous circadian clocks. In mammals, a central pacemaker in the hypothalamic suprachiasmatic nucleus (SCN) integrates external light information to adapt cellular clocks in all tissues and organs to the external light-dark cycle. Together, central and peripheral clocks co-regulate physiological rhythms and functions. In this review, we outline the current knowledge about the routes of communication between the environment, the main pacemakers and the downstream clocks in the body, focusing on what we currently know and what we still need to understand about the communication mechanisms by which centrally and peripherally controlled timing signals coordinate physiological functions and behaviour. We highlight recent findings that shed new light on the internal organization and function of the SCN and neuroendocrine mechanisms mediating clock-to-clock coupling. These findings have implications for our understanding of circadian network entrainment and for potential manipulations of the circadian clock system in therapeutic settings.

## 1. Introduction

Life on earth is subjected to recurrent changes in environmental conditions due to the 24-h rotation of the earth around its axis. The coordination of physiology and behaviour in a 24-h time-dependent manner depends on circadian clocks (from Latin *circa* meaning “about” and *diem* meaning “day”). In mammals, these clocks exist in almost every cell and are synchronized to ensure the adaptation of physiology to external time. Circadian clocks are characterized by their self-sustained rhythms [1] which are temperature compensated [2] and entrainable by periodic external stimuli, so-called *zeitgeber* (German for “time giver”).

Early studies identified the hypothalamic suprachiasmatic nuclei (SCN) as the master central pacemaker, since bilateral lesions of the SCN in rodents result in a complete loss of rhythmic locomotor activity, drinking behaviour, food consumption, hormone release and body temperature [3,4]. The SCN receive photic input from the retina via the retinohypothalamic tract (RHT) but also non-photic input via the intergeniculate leaflet and geniculohypothalamic tract projections [5,6,7]. The neurons in the SCN are organized as a coupled network that conveys the temporal information to other clocks in the brain and peripheral tissues through neural projections and humoral signals finally regulating physiology. The idea of a strict hierarchical organization of the circadian system around the central clock in the SCN has been replaced by the concept of a more “federated” organization since other pacemakers (such as food- and drug-sensitive oscillators) are able to introduce rhythmic circadian output in the absence of the SCN [8,9]. Husse et al. showed that peripheral tissues remain synchronized with the external light-dark cycle even when the SCN clock is absent [10,11]. Exposure to light can immediately affect clock gene expression in non-SCN clocks like the pineal gland or liver, likely through autonomic innervation [12]. Moreover, light can directly activate clock gene expression in adrenal glands independent of photic responses at the level of the central clock [13,14].

Although light has been traditionally considered as the main *zeitgeber* for the circadian system, the time of food intake has a strong impact on liver, kidney, heart and pancreas clocks without affecting the central clock in the SCN [15]. Additionally, scheduled exercise can induce phase shifts in skeleton muscle and lung clocks, probably altering metabolic processes to cope with changing energy demands [16]. Thus, a more federated organization of tissue clocks is necessary to process, integrate and translate environmental signals and ensure the adaptation of rhythmic physiological processes to the environment (see in [17]).

At the molecular level, the circadian clock is based on interlocked transcriptional–translational feedback loops (TTLs) comprised of a set of core clock genes. The positive limb of the mammalian core TTL is formed by the two transcription factors brain and muscle aryl hydrocarbon receptor nuclear translocator-like protein 1 (BMAL1) [18,19,20,21] and circadian locomotor output cycles kaput (CLOCK) [22], which induce the expression of three *period* (*Per1-3*) [23,24] and two *cryptochrome* (*Cry1/2*) [25] genes. In turn, PER and CRY proteins dimerize, translocate to the nuclei and suppress their own transcription [26,27,28] defining the autoregulatory 24 h-loop of the core TTL. Furthermore, BMAL:CLOCK initiate the expression of other genes either directly, via binding to E-box motifs in the promotor of genes e.g., *Reverse-erythroblastosis virus* (*RevErbα/β*) [29] or *albumin d-element-binding protein* (*Dbp*) [30], or indirectly via the oscillation of output genes, so called *clock controlled genes* (*ccgs*). Through these mechanisms the TTLs drive the rhythmic expression of thousands of protein-coding and -noncoding genes. Recent studies show that more than 40% of protein-coding RNAs in mice and 80% in baboons exhibit a ~24-h rhythmic expression in at least one tissue [31,32].

The identity of clock-controlled genes is highly tissue specific as only a minor fraction of oscillating genes is shared across different tissues. In this way, the timing of expression of a particular gene is in phase with the timing of the particular pathway in which this gene is involved [33]. One of the most graphical examples of the former is the rhythmic regulation of liver functions. The liver plays a central role in regulating glucose and lipid metabolism, biosynthesis of serum proteins, cholesterol and xenobiotic biotransformation, among others. The circadian expression of glucose transporters, glucagon and rate-limiting enzymes of glucose utilization is synchronized to peak to prepare for activity [33].

Taken together, the network of cellular clocks needs to process, integrate and translate environmental signals to ensure the adaptation of endogenous physiological rhythms to external time. To this goal, the body’s clocks communicate with each other. The mechanisms of this communication are not fully understood and deciphering systemic clock-to-clock communication remains one of the most important challenges for chronobiologists, since a desynchronized circadian system is believed to contribute to the development of several diseases [34]. The present article will focus on what we already know and what we still need to understand about the communication of different clocks in generating coherent circadian rhythms of physiology and behaviour.

## 2. Central Clock

### 2.1. Local Synchronization Routes

The ability of the SCN to synchronize peripheral clocks rely on a strong neuronal network that transmit time signals to other hypothalamic and extra-hypothalamic nuclei, and connections with the autonomic nervous and the endocrine system. Thus, the SCN entrain other tissue clocks by organizing the rhythms of hormone release and influencing, through the autonomic nervous system, the sensitivity of peripheral tissues to these hormones (reviewed in [35]).

Traditionally, the SCN have been divided into dorsomedial shell and ventrolateral core, based on the retinal innervation patterns and the neuropeptidergic repertoire of the neurons. This feature has been shown in hamsters, mice, rats and humans [36,37,38]. The light signal is transduced by photoreceptive cells in the retina and transmitted to the SCN core along the RHT by glutamate, substance P (SP) and pituitary adenylyl cyclase activating peptide (PACAP). Glutamate activates *N*-methyl-d-aspartate (NMDA) receptors increasing intracellular Ca^2+^ concentrations, which results in the phosphorylation and activation of cyclic adenosine monophosphate (cAMP)-response-element-binding protein (CREB) by protein kinase A (PKA). Activated CREB binds to cAMP response elements (CREs) in *Per1* and *Per2* promoters inducing their transcription. Thus, the rhythmicity of neurons from the SCN core is determined mainly by the photic input. Neurons in the SCN core communicate with the neurons in the SCN shell by several neurotransmitters such as vasoactive intestinal polypeptide (VIP), gastrin-releasing peptide (GRP) and SP [39,40,41]. Cells in the SCN shell exhibit a self-sustained rhythmicity [42] driven by the autoregulatory TTL of clock genes. This property of the SCN was observed when the rhythmic behaviour of dissociate neurons was compared with SCN organotypic cultures [43]. Dissociated SCN neurons exhibit circadian clock gene expression and spontaneous firing rhythms with periods ranging between 20 and 30 hrs. In contrast, SCN neurons in an organotypic culture are synchronized to each other and the period range is much smaller [42]. This suggests that the light signal received by the SCN core pulls the phases of individual neurons in the SCN shell closer together in order to generate a robust and rhythmic output. In turn, SCN shell neurons communicate with hypothalamic and extra-hypothalamic targets by releasing vasopressin (AVP), gamma-Aminobutyric acid (GABA) and diffusible signals (reviewed by [44]).

In a recent paper, core-to-shell and shell-to-core communications have been described in more detail in the murine SCN [45]. The best-studied cell types of the SCN sub-regions are neurons synthesizing AVP in the shell and those expressing VIP in the core region. Moreover, met-enkephalin (ENK)-expressing neurons have been described in the shell and calretinin- (CALR) and gastrin-releasing peptide (GRP)-producing neurons in the core [36]. Current evidence indicates that a complex network organization and communication among different SCN neurons is the key for pacemaker function. The general assumption is that core neurons signal to those of the shell while there is less communication in the reverse direction. However, each individual neuron in the SCN makes as many as 1000 synapses, building a highly specialized network [45,46]. In a recent paper, Park and colleagues performed single-cell transcriptional analysis and revealed novel neuronal phenotypes and interaction networks, providing the basis to fully understand the complex organization of SCN neurons at tissue level [47]. Interestingly, the presence of glia cells in the SCN in an estimated ratio of 3:1 (neurons to glia) adds another level of complexity to the SCN cellular network. SCN astrocytes display antiphasic rhythms of intracellular Ca^2+^ levels compared to neurons. Thus, astrocytes likely play a role in the generation and maintenance of robust SCN circadian oscillations [48,49]. Despite recent advances, the neurochemistry underlying this neuronal–astrocytes network is still not clear.

### 2.2. Central Output

The connections of the SCN with different target organs prepare both the body for upcoming changes in the circadian cycle, and single organs for receiving the hormonal signals associated with these changes. Tracing techniques have shown multiple neuronal projections connecting the SCN with other brain regions. Most of the SCN connections are within the medial hypothalamus where the key cell groups are involved in organizing hormone release and autonomic control [50,51]. These cell groups are located in the medial preoptic area (MPO), the sub-paraventricular area (sub-PVN) and the dorsomedial hypothalamus (DMH). In addition to the ventral and dorsal borders of the PVN, SCN fibres innervate the arcuate nucleus (ARC) and the lateral hypothalamus [52]. Optogenetic induction or suppression of firing in SCN neurons is sufficient to reset the phase and period of the molecular clockwork and alter SCN-dependent entrainment of behavioural rhythms [53].

At the same time, the SCN also receive input from hypothalamic and extra-hypothalamic regions, which allows the adjustment of SCN outputs. Reciprocal neuronal connections mediating this feedback have been described with the ARC [54], the nucleus of the solitary tract (NTS) [55] and the DMH [56]. In a recent paper, Buijs and colleagues tested the SCN–ARC reciprocity by specifically disrupting the connection between these without altering other projections from the SCN to subparaventricular zone-paraventricular nucleus (SPZ-PVN), PVN or DMH. Surprisingly, a specific elimination of SCN–ARC crosstalk results in a complete loss of rhythmicity of one of the main synchronizing hormones, corticosterone, without disrupting SCN clock gene expression [52]. A significant limitation of SCN lesion experiments is that they interrupt neuronal networks with low specificity and may interfere with optic signalling through the nearby RHT. Therefore, in order to dissect the role of the pacemaker clock itself, several models with a specific genetic deletion of the SCN clockwork were generated. These experiments demonstrate that the central pacemaker is important to keep the synchrony of the circadian network in absence of external *zeitgebers* or when conflicting *zeitgeber* signals are received [10,57].

### 2.3. Systemic Synchronization Routes

The influence of the SCN on hormonal secretion and, in turn, the action of these hormones in peripheral tissues is considered as one of the main systemic synchronization routes. However, the SCN also modulate the autonomic nervous system to adapt the sensitivity of peripheral organs to those hormones (reviewed in [35]).

The regulation of circadian glucocorticoid (GC) secretion is one of the best-studied examples of circadian coordination involving a cooperation between the SCN, the autonomic system and adrenocortical clocks [58]. The SCN, via the activation of corticoliberin (CRH) secretion from the paraventricular nucleus of the hypothalamus (PVN), controls the rhythmic release of adrenocorticotropic hormone (ACTH) from the pituitary. ACTH, in turn, stimulates GC production in *zona fasciculata* cells of the adrenal cortex. Via another route involving autonomic pathways, either directly to the adrenal cortex or through connections with the adjacent medulla, the SCN synchronizes adrenal clocks, thereby regulating the sensitivity of the steroidogenic machinery to ACTH stimulation [13,58,59]. Transplantation and knock-down studies suggest that this sensitivity is gated by the adrenal clock. The adrenal gland is more sensitive to ACTH just before the onset of the activity period—as a result, with the same ACTH stimulus the adrenal cortex releases more corticosterone at the beginning of the activity period than at the beginning of the sleep period [58,60]. GC effects are primarily exerted by the glucocorticoid receptor (GR), which is widely expressed throughout the body and within the brain (reviewed by [61]) with the noted exception of the SCN [62]. GCs also bind to and activate the mineralocorticoid receptor (MR) whose expression is restricted to certain tissues [63]. Due to the higher GC affinity to MR, this receptor is tonically activated while GRs are activated only during peaks of ultradian GC pulses or during acute stress responses [64]. GRs act as ligand-activated transcription factors. Upon GC binding, GRs translocate from the cytosol to the nucleus, bind to GRE (glucocorticoid responsive element) DNA motifs in regulatory regions of target genes and modulate transcriptional activity. The daily peak of GCs is synchronized with the need to mobilize energy from tissue stores anticipating the active phase, e.g., promoting gluconeogenesis and glucose release from the liver [65] and fatty acids release from adipose tissues [66]. GR signalling and the molecular clock machinery interact in multiple and reciprocal ways. Hormone-bound GR binds GREs in the promoter regions of several clock genes [67]. Moreover, several clock proteins regulate GR intracellular localization and activity. These multiple bi-directional interactions explain the role of GCs as a major entrainment signal and their role in gating the sensitivity of peripheral tissue to systemic signals across the day (reviewed by [61]).

Melatonin is considered as another key-synchronizing signal, since it exhibits a strong circadian rhythmicity with higher levels during the night (in phase and in anti-phase with GC rhythms in nocturnal and diurnal animals, respectively). In most mammals, rhythmic release of melatonin is regulated by light through sympathetic neuronal connections from the SCN to the pineal gland [3]. Melatonin signal is transduced by G protein-coupled receptors expressed in the SCN, the pituitary and several peripheral organs such as adrenal glands, lung, heart, liver, etc. (reviewed by [68]). For instance, melatonin signalling has been involved in transmitting seasonal day length information to the pars tuberalis, and the circadian regulation of insulin secretion and blood glucose levels [69].

Early studies showed that autonomic nerve activity changes after light exposure while this effect is absent in SCN-lesioned animals [70]. Later, tracing techniques demonstrated that the SCN is connected with several peripheral organs such as adipose tissue, adrenal, heart, liver, ovary, kidney, pancreas, etc. [71,72,73,74,75,76,77]. A combination of tracing techniques and selective denervation also revealed that the SCN is connected with these organs through sympathetic and parasympathetic pre-ganglionic neurons [78]. For instance, leptin, a hormone secreted by adipose tissue, displays a diurnal pattern that is controlled by the SCN through the sympathetic innervation [79]. Although leptin induces phase advances in SCN slices, it may not shift the activity rhythm in vivo but instead potentiate the phase-shifting effect of a light pulse in the late subjective night [80,81]. Electrophysiological experiments demonstrate that the connections of the SCN with neuroendocrine centres in the hypothalamus are physically separated from autonomic connections representing an independent communication route to the periphery [82,83].

## 3. Peripheral Clocks

The systemic synchronizing signals derived from the SCN reach almost all peripheral tissues and there regulate the timing of cellular functions. However, other *zeitgebers* (different from light) and the local clocks play a key role in integrating all the information from the environment to coordinate an appropriate physiological response.

### 3.1. Suprachiasmatic Nucleus (SCN) Independent Entrainment

Food intake is a potent synchronizer for most peripheral clocks including the ones in liver, adipose tissue, muscle, gut and pancreas. A temporal inversion of the normal feeding schedule is able to uncouple the phase of the SCN and peripheral clocks. Thus, while the phase of the SCN clock stays tied to the light–dark cycle, food is the dominant *zeitgeber* for peripheral oscillators [15,84]. One well-described resetting factor in this context is insulin [85]. In response to post-prandial hyperglycaemia, the pancreas releases insulin that is received by peripheral tissues with differential sensitivity. Insulin is able to induce *Per2* expression in target tissues. Liver and adipose tissues are highly responsive to insulin and show a large phase shift of *Per2* expression, while subtle effects were observed in less insulin responsive tissues such as the lung, aorta and submandibular gland [85]. In line with this observation, it was shown that the disruption of the circadian rhythm in insulin secretion by high-fat diet intake is associated with dampened amplitudes of clock gene expression rhythms in the liver [86]. Moreover, similar mechanisms of food-dependent resetting of the peripheral clocks have been described for pancreatic (glucagon), stomach (ghrelin), hepatic (IGF-1) and intestinal (oxyntomodulin) metabolic signals [87,88,89,90].

### 3.2. Local Clocks

The role of local clocks is better assessed in in vitro experiments due to the absence of external systemic timing cues. However, approaches to study the role of a particular tissue clock in the context of a systemic and complex physiological process is complicated by the ubiquitous nature of the clock gene machinery. Before the development of genetic approaches allowing the deletion of clock genes in specific tissues, some transplantation studies provided first insights into the physiological role of local tissue clocks. Later, conditional CRE-loxP-based gene targeting was used to knock out the clock (mainly *Bmal1*, as the only non-redundant clock gene known to date) in numerous peripheral tissues such as liver, pancreas, muscle, heart and adipose tissue [91,92,93,94,95,96]. Reversible CRE systems [97] and, more recently, viral transgene delivery have further been used for tissue-specific deletion of clock function in vivo (reviewed by [98]). By using these tools, we have a relatively good understanding of the role of the adrenal clock in the regulation of the circadian release of GCs. As we already mentioned, light directly activate adrenal clock gene expression via SCN-sympathetic nervous system routes independent from the SCN connection to the hypothalamus–pituitary–adrenal (HPA) axis [13]. Transplantation experiments showed that the adrenal clock gates the sensitivity of the gland to ACTH [58]. However, experiments using adrenal cortex-specific clock deficient mice indicate that the adrenocortical clock itself is dispensable for maintaining normal circadian rhythms of corticosterone release in rhythmic light–dark and constant darkness conditions [99]. A recent paper showed that only during exposure to aberrant light–dark schedules does the adrenal clock play a role stabilizing circadian GC rhythms [100]. Taken together, these results indicate that—rather than regulating baseline circadian rhythms—local clocks may primarily be responsible for integrating different timing signals to produce an appropriate response to external stimuli. 

### 3.3. Integration of Timing Signals

As mentioned before, despite the fact that the SCN serves as the master light-driven synchronizer of the body’s rhythms and clocks, an inversion in the feeding schedule is able to uncouple SCN and peripheral clock rhythms and downstream processes. Le Minh et al. demonstrated that the kinetics of this dissociation between SCN and liver clocks are modulated by GCs [101]. Similarly, when a mouse entrained to a 12 h light/12 h dark (LD) cycle is subjected to phase advance of light by 6 h (a widely used jetlag paradigm) the circadian system will tend to re-entrain all the clocks to the new LD cycle. Kiessling et al. showed that the speed of this re-entrainment is strongly dependent on daily GC rhythms [102]. Years later, Saini et al. developed a system to record the expression of *Bmal1* in the liver using luciferase as a reporter in freely moving mice [103]. In these experiments, they combined timed feeding regimes (restricted to the active or to the resting phase) with SCN lesions. In SCN intact mice, the change of the feeding time from night to day did not immediately impact the phase of the liver clock. In contrast, mice with lesioned SCN showed a rapid shift of the liver clock following the time of food availability [103]. These results indicate that the communication between central and peripheral clocks through systemic signals stabilizes temporal integration when conflicting *zeitgeber* signals (that may be stochastic or intermittent) are received from the environment.

In this context, inter-tissue and inter-cellular synchronization appears to be critical to integrate different timing cues. In a recent study, 24-h metabolic profiles of several mouse tissues were mapped simultaneously showing intra- and inter-tissue metabolic coordination. A comparative analysis showed that within a particular tissue metabolites sharing positive temporal correlation have similar functions or belong to a common pathway while metabolites with a negative temporal correlation represent physiologically incompatible pathways [104]. Importantly, these data provide insights into possible mechanisms regulating inter-tissue temporal coordination and gating of metabolism to specific time windows.

With regard to inter-cellular communication, the presence of similar coupling mechanisms to those present in the SCN has so far not been described for peripheral tissues [43]. Recently, extracellular vesicles (EVs) have been implicated in the circadian synchronization in peripheral tissues [105]. EVs contain and transport molecules such as membrane and cytosolic proteins, mRNAs and non-coding RNAs and their regulatory function depends on receptor–ligand interactions on target cells or direct content delivery after internalization. The cargo of plasmatic EVs isolated from mice exposed to alternating LD cycles—mimicking a night shift schedule—was altered and able to modulate clock gene expression in cultured adipocytes [106]. This evidence constitutes an interesting starting point for further studies.

The intracellular environment is an important regulator of the cell’s clock. One of the first described examples is the regulation of core clock proteins (e.g., the BMAL1:CLOCK dimer) by the oxidized Nicotinamide Adenine Dinucleotide (Phosphate)/reduced Nicotinamide Adenine Dinucleotide (Phosphate) (NAD(P)^+^/NAD(P)H) redox ratio dynamically modulating the dimer’s binding to E-box sequences [107]. Oscillating intracellular NAD^+^ levels will play an additional role in controlling gene expression by activating the deacetylase Sirtuin 1 (SIRT1) that will promote the necessary chromatin remodelling [108]. Moreover, the rhythmic carbohydrate metabolism and mitochondrial oxidative phosphorylation contribute to the oscillation of reactive oxygen species (ROS) levels (reviewed by [109]). In a recent paper, Wible et al. show that a key transcription factor, regulated by oxidative signalling, Nuclear factor (erythroid-derived 2)-related factor 2 (NRF2) is under transcriptional control by CLOCK:BMAL1 contributing to the integration of the cellular rhythmicity to the cellular redox state [110]. A number of other cellular metabolites, e.g., the Adenosine monophosphate/Adenosine triphosphate (AMP/ATP) ratio, have been identified as sensors of the cellular metabolic status. Adenosine monophosphate-dependent activation of the kinase (AMPK), leads to phosphorylation and destabilization of CRY1. Thus, AMPK activation alters the clock phase communicating the energetic state to the local clock machinery [111].

To sum up, the mechanisms underlying communication within the mammalian circadian clock system are complex, highly interconnected and present at all levels—from intracellular signalling to intercellular and systemic coordination (Figure 1 and Table 1). Future studies using targeted genetics and tailored signalling molecules will help us to understand how these mechanisms work together in coordinating an appropriate physiological response to meet environmental demands—and ultimately help in devising manipulative strategies exploiting the clock system communication in therapeutic settings.

## Figures and Tables

**Figure 1 ijms-20-00343-f001:**
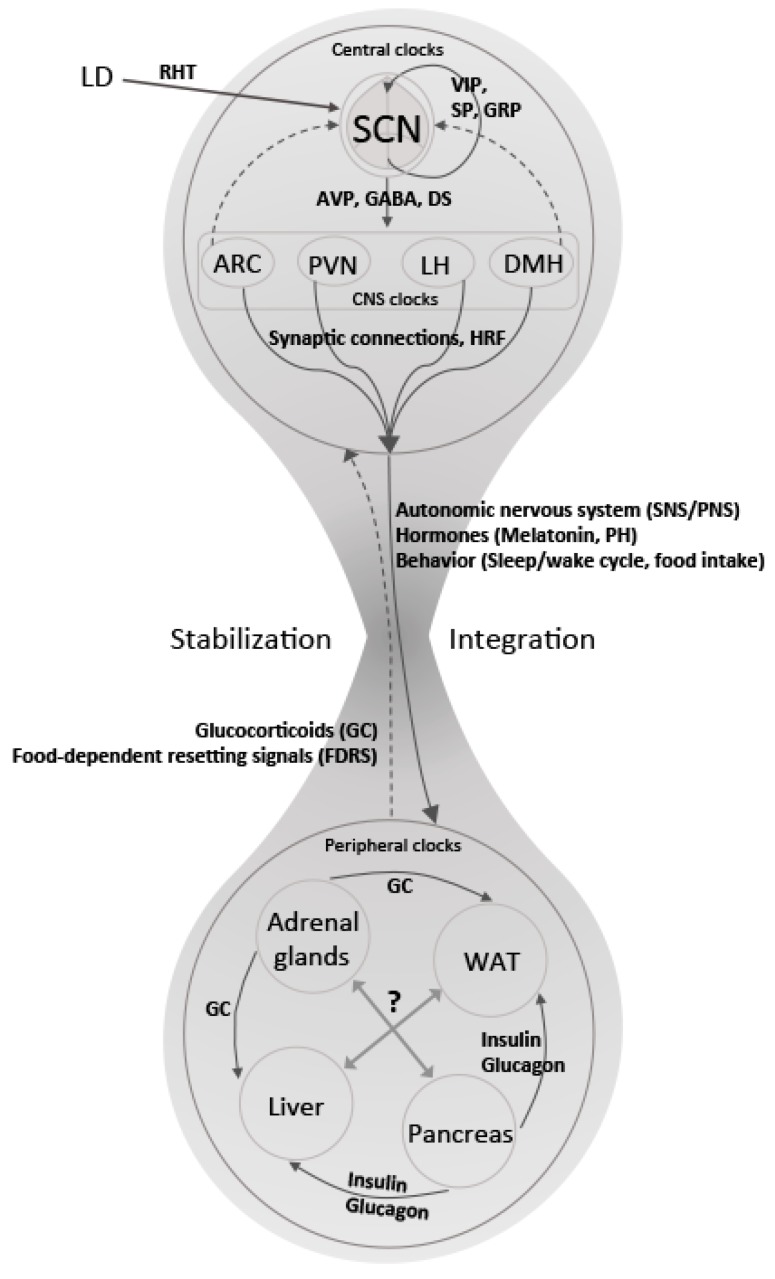
Represents pathways and signals involved in clock-to-clock communication responsible for the integration and stabilization of biological rhythms at central and peripheral levels. The suprachiasmatic nuclei (SCN) receive light information (LD) from the retina as photic input via the retinohypothalamic tract (RHT). The neurons in the SCN are organized as a coupled network of neuronal connections communicating through several neurotransmitters (Vasoactive intestinal peptide (VIP), Substance P (SP), Gastrin releasing peptide (GRP), etc.). The temporal information is then conveyed through Vasopressin (AVP), gamma-aminobutyic acid (GABA) and diffusible signals to other clocks in the brain such as ARC (arcuate nucleus), PVN (paraventricular nucleus), LH (lateral hypothalamus), DMH (dorsomedial hypothalamic nucleus), among others. An integrated response is translated from the brain, through neural projections from the autonomic nervous system and humoral signals, to peripheral tissues (solid arrows). Peripheral clocks receive the time information, communicate with each other and release signals that feed-back (dotted arrows) to the clocks in the brain. The cooperation between central and peripheral clocks results in the stabilization of the rhythms that finally regulate tissue physiology in synchrony with external time.

**Table 1 ijms-20-00343-t001:** Pathways and signals involved in clock-to-clock communication.

Integration	SCN	CNS Clocks	Peripheral Clocks
Intra-cellular	TTL oscillations: cAMP Ca2+ kinase/phosphatase	TTL oscillations: cAMP Ca2+ kinase/phosphatase	TTL oscillations: NAD(P)+/NAD(P)H AMP/ATP Redox balance
Inter-cellular	Core-shell coupling: VIP, GABA, GRP, SP, CALR, ENK, glutamate	Synaptic connections, AVP, GABA	Metabolites (glucose, fatty acids) EVs
Inter-tissue	Photic input (via RHT): Glutamate, PACAP, SP Output to CNS clocks: AVP, GABA, DS	Input from SCN: AVP, GABA, DS Neuroendocrine output: HRF	Insulin, glucagon, GCs, ghrelin, leptin, etc.
Systemic	Behaviour (sleep/wake, food intake) SNS/PNS	SNS/PNS Melatonin PH	GCs FDRS

Body’s clocks communicate with each other to process, integrate and translate environmental signals to adapt physiology and behaviour to the external time. The table shows some of the well-known pathways and signals that are involved in clock-to-clock synchronization at intracellular, intercellular, inter-tissue and systemic levels. TTL: transcriptional translational loop, VIP: vasointestinal polypeptide, GRP: gastrin-releasing peptide, SP: substance P, CALR: calretinin-releasing peptide, ENK: met-enkephalin, RHT: Retino-hypothalamic tract, PACAP: pituitary adenylyl cyclase activating peptide, CNS: Central nervous system, AVP: vasopressin, DS: diffusible signals, SNS: sympathetic nervous system, PNS: parasympathetic nervous system, HRF: hypothalamic releasing factors, PH: pituitary hormones, GCs: glucocorticoids, EVs: extracellular vesicles, FDRS: food-dependent resetting signals (insulin, glucagon, ghrelin, leptin, oxyntomodulin, etc), cAMP: cyclic-Adenosine monophosphate, GABA: gamma-aminobutyic acid, NAD: nicotinamide adenine dinucleotide, AMP/ATP: Adenosine monophosphate/Adenosine triphosphate.

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
