# Peer review of "Mechanisms of Communication in the Mammalian Circadian Timing System"

_ijms, 2019, doi:10.3390/ijms20020343_

Round 1

Reviewer 1 Report

This paper is a comprehensive review focusing on various levels of clock-clock communications, including intra- and inter-cellular communications within nuclei/organs, brain and peripheral organs, and whole-body system via hormones and autonomic nervous system. Early and recent papers are well referred and combined to follow the research trend. There are several concerns described below. 

1.    Line 33: In addition to the direct input via RHT, the indirect input via intergeniculate leaflet and geniculohypothalamic tract projects to the SCN and adjusts photic and non-photic entrainment. This route should be also mentioned. 

2.    Line 40: References for light-induced activation of clock gene expression is limited to the adrenal gland (ref 7,8). The references for other tissues such as the pineal gland or liver can be added, even though other tissues may be under control of the SCN (e.g., Cailotto et al., PLoS One, 4:e5650, 2009).

3.    Line 87-: The light signal transduction pathway including acute induction of Per1 and Per2 expression and phase shifts occurs when the light was given during subjective night, but not subjective day. This should be clarified. 

4.    Line 138-139: Clarify “SPZ-PVN”. 

5.    Line 139-140: It should be mentioned that expression rhythm of clock genes in the SCN was intact even when SCN-ARC was eliminated and corticosterone rhythm was disrupted (ref 46).

6.    Line 179: The role of melatonin in clock synchronization is referred for the regulation of insulin and glucose. However, the best studied function is the transmission of day-length information to controlling center of photoperiodic response, the pars tubelalis, to coordinate seasonal physiology via clock genes and other genes. This should be mentioned in the section. 

7.    Section 2.3 Systemic synchronization routes: Authors may also include leptin signaling as another route, because leptin secretion is regulated by SCN and modulates clock genes expression and phase shifts.

8.    Line 212-214: A recent study reported the resetting of the liver clock by glucagon and IGF-1 (Ikeda et al., EBioMedicine, 28:210-224, 2018), which may be also referred. 

9.    Reference list: Capitalized and non-capitalized title are mixed. 

Author Response

We very much thank reviewer 1 for his/her thoughtful comments on our manuscript. Please find a point-by-point rebuttal below.

Reviewer’s #1 comments to the authors:

This paper is a comprehensive review focusing on various levels of clock-clock communications, including intra­ and inter­cellular communications within nuclei/organs, brain and peripheral organs, and whole­body system via hormones and autonomic nervous system. Early and recent papers are well referred and combined to follow the research trend. 

There are several concerns described below. 

1. Line 33: In addition to the direct input via RHT, the indirect input via intergeniculate leaflet and geniculohypothalamic tract projects to the SCN and adjusts photic and non­photic entrainment. This route should be also mentioned. 

Reply: We thank the reviewer for this comment. In the revised version, we have modified the sentence to “The SCN receive photic input from the retina via the retinohypothalamic tract (RHT) but also non-photic input via the intergeniculate leaflet and geniculohypothalamic tract projections.“ (Page 1)

2. Line 40: References for light­induced activation of clock gene expression is limited to the adrenal gland (ref 7,8). The references for other tissues such as the pineal gland or liver can be added, even though other tissues may be under control of the SCN (e.g., Cailotto et al., PLoS One, 4:e5650, 2009).

Reply: We appreciate this comment. In the revised version we now state: “Exposure to light can immediately affect clock gene expression in non-SCN clocks like pineal gland or liver, probably through autonomic innervation. Moreover, light can directly activate clock gene expression in adrenal glands independent of photic responses at the level of the central clock.“ (Page 1 & 2).

3. Line 87­: The light signal transduction pathway including acute induction of Per1 and Per2 expression and phase shifts occurs when the light was given during subjective night, but not subjective day. This should be clarified. 

Reply: We agree with the reviewer. Although the results were obtained using light pulses/administering neurotransmitters in the subjective night, the aim of this paragraph was to describe, in a simplified way, the molecular pathways through which light can principally affect Per expression in the SCN. Therefore, we decided not to change the text.

 4. Line 138­139: Clarify “SPZ­PVN”. 

Reply: We apologize for the mistake. We add the appropriate abreviation in the revised version. (Page 3 line 221)

 5. Line 139­140: It should be mentioned that expression rhythm of clock genes in the SCN was intact even when SCN­ARC was eliminated and corticosterone rhythm was disrupted (ref 46).

Reply: We rephrased this sentence in the revised version according to the reference mentioned by the reviewer. Now it reads: “ Surprisingly, a specific elimination of SCN-ARC crosstalk results in a complete loss of rhythmicity of one of the main synchronizing hormones, corticosterone, without disrupting SCN clock gene expression”. (Page 4)

6. Line 179: The role of melatonin in clock synchronization is referred for the regulation of insulin and glucose. However, the best studied function is the transmission of day­length information to controlling center of photoperiodic response, the pars tuberalis, to coordinate seasonal physiology via clock genes and other genes. This should be mentioned in the section. 

Reply:  We acknowledge and agree with this comment. We added this information: “For instance, melatonin signaling has been involved in transmitting seasonal day­length information to the pars tuberalis, and the circadian regulation of insulin secretion and blood glucose levels [70].” (page 4)  

7.    Section 2.3 Systemic synchronization routes: Authors may also include leptin signaling as another route, because leptin secretion is regulated by SCN and modulates clock genes expression and phase shifts.

Reply: We agree and acknowledge the comment. We included leptin as a systemic sysnchronizing route. Now it reads: “For instance, leptin, a hormone secreted by adipose tissue, displays a diurnal pattern that is controlled by the SCN through the sympathetic innervation. Although leptin induces phase advances in SCN slices it may not shift the activity rhythm in vivo but instead potentiate the phase-shifting effect of a light pulse in the late subjective night”. (Page 5)

8. Line 212­214: A recent study reported the resetting of the liver clock by glucagon and IGF­1 (Ikeda et al., EBioMedicine, 28:210­224, 2018), which may be also referred. 

Reply: The reference was added as suggested.

9. Reference list: Capitalized and non­capitalized title are mixed. 

Reply: The format of the references in the reference list was carefully revised.

Reviewer 2 Report

Reviewer Comments:

The authors do a nice job of highlighting an important area of current research in the field of biological clocks- mechanisms that mediate the integration of multiple circadian timing mechanisms in the organism.  The main value of this review is that the level of focus is on actual mechanisms at the physiological and cellular level that we need to understand in order to fill in the black boxes in the more formal functional models that have established the existence of multiple central and peripheral circadian pacemakers and zeitgebers and their complex interactions.  As the authors say, it is still necessary to figure out the proximate mechanisms that connect the circadian timing mechanisms to the numerous driven rhythms at the physiological and metabolic endpoints, and this is one of the current major areas of expansion in biological clocks research that is ripe for progress and bringing new disciplines into the field.

In short, the authors should emphasize their review of current research on mechanisms that mediate circadian system organization and omit the long-obsolete concept of a single master light-driven pacemaker in the SCN governing all circadian rhythms as a rationale for their review. The argument that their review reveals new insights into circadian mechanisms at this level makes their rationale sound obsolete and naïve since the concept of multiple central circadian pacemakers and zeitgebers and peripheral oscillators functioning as a circadian system has been standard dogma in the field for well over a decade, and the relevance of this review is in its timely focus on the integrative mechanisms mediating circadian system function, which is still poorly understood, rather than shedding any new light on the circadian system idea, which is already well established.

Other than that, the nuts and bolts of the author's review provide a useful focus on complex cellular and physiological mechanisms that connect circadian system timing elements to each other and to driven rhythms. Their discussion of the SCN shell and core structure and function, for instance, is very useful for circadian researchers who are not neuroscientists, and their overall emphasis on neuroendocrine mechanisms in the context of circadian organization is similarly valuable.

Overall, I think the paper provides a useful and relevant review of circadian system mechanisms in mammals, but before publication the rationale should be revised to recognize that the circadian system concept is well established, and this review is designed to further our understanding of the mechanisms mediating system-wide integration.

Specific comments:

 1. Abstract:  After the first two sentences (line 11), the authors should note that additional circadian pacemakers interact with the SCN and co-regulate physiological mechanisms, such as the food-entrainable pacemaker, redox oscillator and methamphetamine sensitive oscillator.

 The fourth sentence (lines 15-16) should be revised to read something like "We highlight recent findings that shed new light on the internal structure and function of the SCN, and neuroendocrine mechanisms mediating clock-to-clock coupling."  

 The current text "….traditional models predicting that (i) SCN entrainment is necessary to adapt body clocks to the external light-dark cycle and (ii)" should be deleted, and so should "…is restricted to the SCN pacemaker." in the same sentence.

2. Introduction.  The last sentence of the first paragraph (line 28): change "by external stimuli" to "by periodic external stimuli" and change "so called zeitgeber" to "the so-called zeitgebers".

 Lines 30-31: change "…master central pacemaker. Bilateral lesions…" to "…master central circadian pacemaker, since bilateral lesions…"

Line 36: change   "Recently, this idea…" to "Since this idea…."

Line 37: change "…has been challenged…" to "…has been replaced by the concept of a circadian system…." and here you might cite one or more of the earlier papers reviewing the circadian system concept such as Mohawk et al. (2012) Annu Rev Neurosci 35:445-62, or Dibner et al. (2010) Annu Rev Physiol. 72:517-49).

Line 41: change "Despite light has been…" to "Although light has been…"

Line 42: change "…time of food intake may have a strong impact…" to "…time of food intake has a strong impact…"

Line 45: change "…organization of tissue clocks seems to be necessary…" to "…organization of tissue clocks is necessary…."

Lines 69-76: The authors concluding paragraph for the Introduction perfectly sums up the value of this review, and the revisions suggested above to the abstract and introduction point to this paragraph, which moves the field forward in a useful direction,  rather than re-hashing older arguments about the paradigm shift from "the" SCN master pacemaker to a more complex circadian system.

Section 3.3 Integration of timing signals: Line 238. Change "…master synchronizer…" to "…master light-driven synchronizer…."

Lines 261-262: change "…similar coupling mechanisms as the ones present in the SCN…" to "…similar coupling mechanisms to those present in the SCN…"

Line 263-264: This sentence should be followed by one or more references since it is making an important declaration.

Author Response

We also very much thank reviewer 2 for his/her thoughtful comments on our manuscript. Please find a point-by-point rebuttal below.

Reviewer’s #2 general comments to the authors:

The authors do a nice job of highlighting an important area of current research in the field of biological clocks mechanisms that mediate the integration of multiple circadian timing mechanisms in the organism. The main value of this review is that the level of focus is on actual mechanisms at the physiological and cellular level that we need to understand in order to fill in the black boxes in the more formal functional models that have established the existence of multiple central and peripheral circadian pacemakers and zeitgebers and their complex interactions. As the authors say, it is still necessary to figure out the proximate mechanisms that connect the circadian timing mechanisms to the numerous driven rhythms at the physiological and metabolic endpoints, and this is one of the current major areas of expansion in biological clocks research that is ripe for progress and bringing new disciplines into the field. In short, the authors should emphasize their review of current research on mechanisms that mediate circadian system organization and omit the longobsolete concept of a single master light-driven pacemaker in the SCN governing all circadian rhythms as a rationale for their review. The argument that their review reveals new insights into circadian mechanisms at this level makes their rationale sound obsolete and naïve since the concept of multiple central circadian pacemakers and zeitgebers and peripheral oscillators functioning as a circadian system has been standard dogma in the field for well over a decade, and the relevance of this review is in its timely focus on the integrative mechanisms mediating circadian system function, which is still poorly understood, rather than shedding any new light on the circadian system idea, which is already well established. Other than that, the nuts and bolts of the author's review provide a useful focus on complex cellular and physiological mechanisms that connect circadian system timing elements to each other and to driven rhythms. Their discussion of the SCN shell and core structure and function, for instance, is very useful for circadian researchers who are not neuroscientists, and their overall emphasis on neuroendocrine mechanisms in the context of circadian organization is similarly valuable.

Overall, I think the paper provides a useful and relevant review of circadian system mechanisms in mammals, but before publication the rationale should be revised to recognize that the circadian system concept is well established, and this review is designed to further our understanding of the mechanisms mediating system­wide integration. 

Reply: We agree and acknowledge the reviewer’s comment on the rationale of our review. Therefore, we made several changes to strengthen the focus of our article on the integration mechanisms. (see, e.g., the amended abstract)

Specific comments:

1. Abstract: After the first two sentences (line 11), the authors should note that additional circadian pacemakers interact with the SCN and coregulate physiological mechanisms, such as the food­entrainable pacemaker, redox oscillator and methamphetamine sensitive oscillator.

The fourth sentence (lines 1516) should be revised to read something like "We highlight recent findings that shed new light on the internal structure and function of the SCN, and neuroendocrine mechanisms mediating clock to clock coupling."

The current text "….traditional models predicting that (i) SCN entrainment is necessary to adapt body clocks to the external lightdark cycle and (ii)" should be deleted, and so should "…is restricted to the SCN pacemaker." in the same sentence.

Reply: We appreciate the reviewer’s comment on the abstract. We have made several changes to repframe the scope of the review.

2. Introduction.  The last sentence of the first paragraph (line 28): change "by external stimuli" to "by periodic external stimuli" and change "so called zeitgeber" to "the so­called zeitgebers".

Lines 3031: change "…master central pacemaker. Bilateral lesions…" to "…master central circadian pacemaker, since bilateral lesions…"   Line 36: change   "Recently, this idea…" to "Since this idea…."

Line 37: change "…has been challenged…" to "…has been replaced by the concept of a circadian system…." and here you might cite one or more of the earlier papers reviewing the circadian system concept such as Mohawk et al. (2012) Annu Rev Neurosci 35:44562, or Dibner et al. (2010) Annu Rev Physiol. 72:51749).   

Line 41: change "Despite light has been…" to "Although light has been…"

Line 42: change "…time of food intake may have a strong impact…" to "… time of food intake has a strong impact…"

Line 45: change "…organization of tissue clocks seems to be necessary…" to "…organization of tissue clocks is necessary…."   

Lines 6976: The authors concluding paragraph for the Introduction perfectly sums up the value of this review, and the revisions suggested above to the abstract and introduction point to this paragraph, which moves the field forward in a useful direction, rather than re­hashing older arguments about the paradigm shift from "the" SCN master pacemaker to a more complex circadian system.

Reply: We thank the reviewer for his/her valuable comments on the introduction. We have made the changes suggested and several others that we consider appropriate according to the comments. We also added the suggested references. 

Section 3.3 Integration of timing signals: Line 238. Change "…master synchronizer…" to "…master light­driven synchronizer…."

  Lines 261262: change "…similar coupling mechanisms as the ones present in the SCN…" to "…similar coupling mechanisms to those present in the SCN…" 

Reply: We have corrected these sentences as suggested.

Line 263264: This sentence should be followed by one or more references since it is making an important declaration.

Reply: We apologize for this mistake. We decided to remove the sentence since, at this moment, there is no strong experimental evidence of intercellular coupling in peripheral tissues.

Reviewer 3 Report

This is a well written review paper about entrainment of mammalian circadian clock in each cell. It will make a better review paper if the authors add their opinion about the missing links between two entrainment mechanism. It is acceptable without adding any other information.

Author Response

We very much thank reviewer 3 for his/her positive evaluation of our manuscript. We hope that the changes made to the revision have further improved the paper.